# Rapid Drop in Coronary Heart Disease Mortality in Czech Male Population—What Was Actually behind It?

**DOI:** 10.3390/biomedicines10112871

**Published:** 2022-11-09

**Authors:** Rudolf Poledne, Anna Kralova, Hana Bartuskova, Karel Paukner, Sona Kauerova, Jiri Fronek, Vera Lanska, Ivana Kralova Lesna

**Affiliations:** 1Laboratory for Atherosclerosis Research, Institute for Clinical and Experimental Medicine, 140 21 Prague, Czech Republic; 2Transplantation Surgery Department, Institute for Clinical and Experimental Medicine, 140 21 Prague, Czech Republic; 3Department of Informatics, Institute for Clinical and Experimental Medicine, 140 21 Prague, Czech Republic; 4Anesthesiology, Resuscitation and Intensive Care Unit, Military University Hospital, 140 21 Prague, Czech Republic

**Keywords:** diet, economy, inflammation, macrophages, coronary heart disease mortality, cholesterol, n-3 fatty acids

## Abstract

The high mortality of coronary heart disease (CHD) among Czech men—one of the highest worldwide—began to decline in 1991 soon after the abolition of government subsidies to all foodstuffs rich in animal fat. As participants in the WHO MONICA Project, we were able to analyze the CHD risk factors just before and after this major economic change. We had previously documented that the originally subsidized prices decreased animal fat consumption and consequently non-HDL cholesterol concentrations in the population. By the early 1990s, no progress had been made in the treatment of acute myocardial infarction, statins were unavailable as was not the currently more effective antihypertensive therapy. Our recent research proved a close relationship between cholesterolemia and proinflammatory macrophages in adipose tissue and accelerated macrophage polarization with increased palmitate and palmitoleate contents in cell membrane phospholipids. By contrast, the proportion of proinflammatory macrophages decreases with increasing presence of n-3 fatty acids in the cell membrane. The combination of non-HDL cholesterol drop and a decreased proportion of proinflammatory macrophages due to replacement of alimentary fat decreased CHD mortality immediately.

## 1. Introduction

Coronary heart disease (CHD) mortality has been decreasing continuously in the industrialized nations since the mid-1960s, first in the USA [1], followed by similar changes in Australia and Western Europe. The very beginning of this encouraging development in the USA and the United Kingdom is difficult to pinpoint as no definite changes in the risk factors or treatment of acute coronary syndrome heralding a turn of the tide could be identified [2]. The subsequent trajectory of CHD mortality in the 1970s and 1980s is believed to have been dependent on several different effects. It was essentially unknown at that time whether or not the decline was occurring because of the lower incidence of new cases (beneficial changes in cardiovascular health-related habits, number, and severity of risk factors, and preventive medicine) or substantial improvements in the medical management of acute events [2].

The late 20th century brought completely new perceptions of the importance of lifestyle change, primary prevention, and treatment of established coronary heart disease [3]. With the introduction of the IMPACT model, the effects of acute treatment on the clinical complications of atherosclerosis in the USA were described and compared with current practices in New Zealand and a few European countries [4]. In the 1990s, estimates of the relative importance of risk factor modification in Europe varied from 44% (in The Netherlands) to 55% (in Scotland), with the exception of 76% in Finland [4]. An updated version of the IMPACT model was used to analyze the declining rates of CHD mortality in Sweden [5]. The greatest contributor to this decrease was a reduction in major risk factors which accounted for 55%, whereas only 36% was believed to have been due to all other types of management of acute complications of CHD. The effect of diet on reducing cholesterol levels was estimated at only 40% of the total, representing the most important factor of all those analyzed.

New substantial information about progression of coronary atherosclerosis has been published within the last two decades, documenting the interplay of intravascular levels of cholesterol-containing lipoprotein particles [6] and immune processes in the subendothelial space of coronary arteries. Data documenting the presence and importance of monocytes/macrophages in the atherosclerotic plaque, both in man and experimental models, are available in reviews [7,8]. Although the effect of monocytes/macrophages in atherogenesis occurs within the subendothelial space of arteries, it may also manifest itself through the systemic proinflammatory status [9,10] induced by adipose tissue enlargement. Adipose tissue enlargement induces proinflammatory changes and increasing amounts of resident-tissue macrophages [11]; however, this relationship is not simple [12].

We have documented earlier [13,14] that the proportion of proinflammatory macrophages in human adipose tissue correlated closely with non-HDL cholesterol levels. The pathophysiology of coronary atherosclerosis is currently perceived as a combination of shear stress in the coronary arteries leading to changes in the endothelium [15] that induce adhesion of monocytes and their migration to the subendothelium and increasing penetration of cholesterol-containing lipoproteins [7,16,17]. This synergy accelerates atherogenesis, and it also stabilizes pre-existing plaques [18] when the risk is decreasing. In the present paper, we combine our earlier epidemiologic, biochemical [19], anthropometric, as well as economic data with recent experimental data related to human adipose tissue inflammation. Our goal was to take advantage of a major socioeconomic change followed by a dramatic drop in CHD mortality and our participation in the well-designed population-based WHO MONICA study to be merged with our recent results.

## 2. Materials and Methods

### 2.1. Epidemiological Data

Data related to CHD mortality as well as CHD treatment were provided by the Institute of Health Information and Statistic of the Czech Republic based on individual death certificates. Biochemical and anthropological data of the Czech general population were obtained from a representative 1% random population stratified sample of residents from six regions of the Czech Republic in the 1st and 2nd WHO MONICA surveys conducted in 1988 and 1992, respectively. In these surveys, 2570 and 2768 individuals aged 25–64 years were examined with response rates of 81% and 73%, respectively.

### 2.2. Biochemistry

Total cholesterol, triglyceride, and HDL cholesterol fractions of living kidney donors (LKDs) were determined from fasting blood samples obtained immediately before surgery (prior to anesthesia) using an enzymatic method (Hoffmann-LaRoche, Basel, Switzerland). High-sensitivity C-reactive protein was measured by immunoturbidimetric assay using a Cobas Mira Plus Autoanalyzer (Hoffmann-LaRoche, Basel, Switzerland). All parameters were analyzed in a lipid laboratory participating in the CDC Atlanta, GA, USA control system and serving as a control laboratory within the WHO MONICA project.

### 2.3. Analysis of Adipose Tissue

All 154 individuals (enrolled between July 2014 and December 2020) were fully informed about the process of kidney donation and transplantation. All participants signed informed consent forms in relation to adipose tissue sampling. Samples of VAT were obtained intraoperatively to be immediately cooled and transferred to the laboratory. The method of adipose tissue processing has been described in detail earlier [20]. Stromal vascular fraction (SVF) was isolated and analyzed after cleaning, dissecting, and disintegrating by collagenase (2 mg/L) for 15 min (37 °C). Monoclonal antibodies and fluorochromes (CD14, phycoerythrin-cyanine, CD16, phycoerythrin-Texas Red X (ECD), CD36, and fluorescein isothiocyanate (FITC)) were used to distinguish different subsets of viable monocytes/macrophages. 

Based on our results published earlier [21], we defined cells with a combined phenotype CD14+ and CD16+ with high phagocytic receptor CD36 expression as normally stimulated M1 proinflammatory macrophages (PIMs), whereas those CD14+ with no CD16 expression and low CD36 expression represented M2 anti-inflammatory macrophages. However, we are well aware that this approach is an oversimplification of the complex in vivo situation where several transient phenotypes not followed in our study may exist and their role should be considered. In the present study, these minor transition fractions represented, altogether, 15 ± 2.3% of the total of macrophages within the adipose tissue of LKDs, varying between 6% and 35%.

### 2.4. Analysis of Fatty Acid Composition

The extraction, separation, and methylation of adipose tissue phospholipids have been described in detail earlier [22]. Total lipids were extracted with dichlormethane:methanol using a modified Folch method and phospholipids were isolated by thin-layer chromatography. The fatty acids in phospholipids of the whole adipose tissue were converted to methyl esters and separated by gas chromatography using a Hewlett-Packard GC system with a flame ionization detector and a carbowax-fused silica capillary column [23]. Individual peaks of FA methyl esters were identified by comparing retention times with those of authentic standards (mix of standard FAs, Restek Corporation, Bellefonte, PA, USA). The relationships of the eight main FAs in tissue membrane phospholipids with the macrophage phenotype were assessed (*n* = 43). The other minor FAs (<2%) were pooled to saturated (SAFA), monounsaturated (MUFA), and n-3 and n-6 polyunsaturated fatty acids (PUFA), accordingly.

### 2.5. Statistics

Our data are presented as means with standard deviations for continuous variables. The correlations of the proportion of PIMs to non-HDL cholesterol and different FAs were documented using the coefficient of correlation *r*, which was calculated with the Pearson method, including the *p*-value. All tests were two-tailed and the level of significance was set at 0.05. Statistical analyses were performed using Prism 6 (GraphPad Software, Inc., La Jolla, CA, USA).

## 3. Results

### 3.1. Epidemiological, Economical, and Experimental Data

During the 1960s and 1970s, CHD mortality in the Czech general population was still increasing in contrast to the progressive downward trend seen in the industrialized Western countries. This trend in CHD mortality ranked the Czech Republic among countries with the highest CHD-related death rates worldwide. During the early 1980s, CHD mortality in men (Figure 1) somewhat plateaued to be followed by a significant downward trend between 1985 and 1988. An unexpected peak occurred in 1990 (with a similar peak observed in Poland [24]) with a prominent downward trend continuing until 2010. The CHD mortality rates in Czech women displayed a similar pattern with only modest differences. Similar to other countries, the decrease was driven by a combination of medical and lifestyle modifications [4]. The only exception in the Czech population was seen in the 1991–1994 period, when no changes in the health care service were introduced and the proportion of gross national product to health service budget was rather low, varying between 3.7 and 4.2%; hence, the dramatic decline in CHD mortality since 1991 should be attributed solely to lifestyle modifications.

#### 3.1.1. Treatment of Atherosclerosis Risk Factors and Acute Myocardial Infarction

Based on data obtained from the Institute of Health Information and Statistics of the Czech Republic and those released by the Czech Society of Cardiology, acute treatment of CHD remained rather unaltered during this period [25]. Primary and secondary prevention of CHD did not change either, as no statins were available at that time and use of novel antihypertensive agents was limited. This is best illustrated by the percentages of gross national product spent on health service, i.e., 3.72, 3.35, and 3.42% in 1991, 1992, and 1993 (similar to the 1980s), changing substantially, in 1994, with the advent of health insurance companies. The fees paid by those insured (almost 100%) were shifted directly to health insurance companies and not indirectly as part of the state budget assigned to cover health service costs, whereby, the proportion of funds allocated to the health care service increased to 5.2% in 1994, and thereafter, ranged between 5 and 6%.

#### 3.1.2. Economic Change and Atherosclerosis Risk Factors

On the contrary, a major economic change came into effect on 1 January 1991 in the form of universal price liberalization. The Czech government decided to deregulate all prices, not only those of steel or corn, to realistically reflect those on the international market while also stopping subsidies to meat, meat products, milk, and dairy products (up to 50% of the total price). As a result, and to give an example, the price of butter doubled and, consequently, its consumption decreased from 10.2 kg/capita per year in 1989 to 5.3 kg in 1992. Generally, the economic change and abolition of subsidies to animal products placed immediate pressure on the budgets of the majority of families. The concentration of non-HDL cholesterol dropped by 14% in the population sample and the distribution curve of cholesterol concentration was shifted to the lower concentration (Figure 2) between 1988 and 1992 as a consequence of the changes in FA consumption. Understandably, after a positive dietary change, there was also a slight decrease in HDL cholesterol concentration. The ratio of total cholesterol/HDL concentrations significantly decreased from 5.01 to 4.66, documenting a substantial decrease in CHD risk. Unlike the slightly decreased consumption of SAFA, an increase in n-3 FA consumption appeared, mainly by increased consumption of α-linolenate due to replacement of butter by soft margarine and rapeseed oil.

#### 3.1.3. ProInflammatory Macrophages in Visceral Adipose Tissue and Cholesterolemia

Essentially, we can divide the bulk of adipose tissue macrophages into three basic phenotypes. The proportions of PIMs in human visceral adipose tissue range between 20 and 60% of total adipose tissue macrophages characterized as CD14+, CD16+, and CD36^high^. Correspondingly, anti-inflammatory adipose tissue macrophages with the CD14+CD16-CD36^low^ phenotype shows a mirror-like pattern. In addition to these two main subfractions, adipose tissue contains several intermediate subfractions representing an average 15 ± 2%. Our preliminary data [13] documented a significant positive correlation between the proportion of PIMs and non-HDL cholesterol levels in a relatively small number of LKDs. We, herein, present this correlation in 154 LKDs (Figure 3) [26]. On the one hand, the highly significant correlation demonstrated that, in individuals with low non-HDL cholesterol levels, the proportion of PIMs was about 20%. On the other hand, the proportion of PIMs in the visceral adipose tissue of hypercholesterolemic individuals (non-HDL cholesterol levels >4.5 mmol/L) was in the range of about 50–60%, i.e., almost twice to three times as high. When the proportion of PIMs was related to the level of non-HDL cholesterol in our entire group of LKDs, a very highly positive and significant correlation (*p* < 0.0001) with an r2 value of 0.1288 was found. As is evident from the shift of the curve of distribution of cholesterol concentrations, the decrease in the proportion of PIMs occurred in the whole population sample (Figure 3). The change in the proportion of PIMs was estimated at 18% for every 1 mmol of increase in non-HDL cholesterol levels.

Looking at the relationship, we might propose that the documented shift in non-HDL cholesterol levels in the population in the early 1890s implies a decrease in the proportion of PIMs in visceral adipose tissue.

#### 3.1.4. The Proportion of Proinflammatory Adipose Tissue Macrophages and Fatty Acid Composition

Finally, we analyzed the spectra of FAs in the adipose tissue phospholipids in our group of LKDs. The proportion of PIMs correlated significantly with several FAs in the adipose tissue phospholipids analyzed. Principally, the proportion of PIMs correlated positively (Figure 4) with the levels of palmitate and (its desaturated product) palmitoleate, while not correlating with the levels of monounsaturated FAs. The correlation between the proportion of PIMs and palmitoleate content being the closest among all FAs. On the contrary, the proportion of PIMs correlated negatively with content of both α-linoleate and eicosopentaenate (EPA). A significant inverse correlation of all n-3 FAs to α-linolenic acid was likewise documented. In addition, a very significant inverse correlation between the proportion of PIMs and the n-3/n-6 fatty acid ratio was found. Although the content of EPA in human adipose tissue is rather low, this polyunsaturated FA also correlated inversely highly significantly with the proportion of PIMs.

In summary, the effect of the FA spectrum in human adipose tissue on the proinflammatory status and polarization of tissue-resident macrophages was substantial. As the presence of palmitate and palmitoleate in the phospholipid fraction of adipose tissue increased, the character of adipose tissue became more proinflammatory. On the contrary, with increasing presence of α-linoleate and total n-3 FAs, the proportion of PIMs tended to decline. A very potent effect was also documented for EPA.

## 4. Discussion

### 4.1. CHD Mortality Changes and Cholesterol

The impressive downward trajectory of CHD mortality in Czech males for two decades has extended their life expectancy by almost 7 years. Two decades ago, we suggested [19] that this decrease in CHD mortality rates may have been initially due to the elimination of the long-term program of subsidies to animal products (believed to be healthy after the end of World War 2) and subsequent decrease in the levels of atherogenic lipoproteins. This statement was based on the quite simplified principles of the pathophysiology of hyperlipoproteinemia and atherosclerosis, disregarding as it did the interplay between atherogenic lipoproteins and inflammation in the formation of atherosclerotic plaques and their stability [27].

The very beginning of this drop was likely induced only by the change of diet at that time due to economic pressures on the majority of family budgets. The decreased consumption of SAFAs (contained in the foodstuffs subsidized up to that time) and increase in PUFAs (contained in the originally non-subsidized foodstuffs) reflected these price changes. We took advantage of the participation of our institutions in the large epidemiological WHO MONICA study and related parameters were analyzed in a representative population sample. The first MONICA survey was conducted just before the economic changes and the second survey was conducted just after the economic changes, making the drop in non-HDL cholesterol concentrations as well as the shift in cholesterolemia in the whole population sample clearly evident (Figure 2).

An older concept of atherogenesis already assumed qualitative changes in the endothelium of the inner arterial lining, which increases the rate of inflow of cholesterol-carrying lipoproteins (necessary for the normal functioning of smooth muscle cells) together with inflow of circulating monocytes. The presence of cholesterol-carrying particles in the subendothelial space higher than the physiological requirement of the arterial wall cells accelerates scavenging of these particles in local macrophages by specific receptors. On the one hand, cholesterol accumulation (when reverse cholesterol transport becomes inadequate) leads to foam cell production and sets off the process of atherogenesis. On the other hand, a decrease in cholesterol inflow to the artery may start a reversal of atherosclerosis and reduce the likelihood of atherosclerosis-related complications [28]. In addition, there may also be decreased transport of cholesterol molecules to the lipid rafts in circulating monocytes [26]. The documented decrease in non-HDL cholesterol levels and the shift of the general population to lower cholesterol levels, thus, resulted in lowering CVD mortality.

### 4.2. CHD and Inflammation

At the same time, a substantial decrease in the prevalence of obesity from 25 to 20% (BMI > 30) was demonstrated in the MONICA analysis. Adipose tissue hypertrophy is connected with substantial proinflammatory changes, so one may assume that systemic subclinical inflammation also decreased. Systemic inflammation may have been influenced by the diet change per se, as demonstrated previously [28]. This effect became obvious as early as the first year after the economic change and the consequences of a reduction in adipose tissue volume probably stabilized the incidence of coronary atherosclerosis and decreased the incidence of acute coronary syndrome. Hanson’s laboratory (among others) focused their attention on the crucial importance of monocytes in the pathogenesis of atherosclerosis [7]. These cells are active in monocyte/endothelial adhesion and produce numerous protein-attracting cells, which promote the process of atherogenesis. In addition, macrophages play a dominant role in the regulation of systemic inflammation born in adipose tissue [29]. It is widely acknowledged that the number of macrophages and also the proportions of their different phenotypes play a role in local inflammatory change [30]. We have already documented direct links among macrophage phenotypes in the adipose tissue, the different cardiovascular disease predictors [31], and diet [14].

Therefore, we suggest that our data demonstrate an immediate effect of dietary modifications on coronary plaque stabilization and, consequently, reduced likelihood of CHD-related clinical complications and reduced CHD mortality, in agreement with the relevant data [32]. Based on our previous results [14], the cholesterol molecule as well as a discrepancy of n-3 FAs and SAFAs in phospholipids of the adipose tissue cell membrane (Figure 4) affect proinflammatory macrophage polarization, systemic inflammation [16], and the process of atherogenesis [27,33].

### 4.3. Combined Effect

Although statin treatment has been shown to be most effective in decreasing the incidence and severity of atherosclerosis-related clinical complications, there is still a proportion of individuals dying from acute myocardial infarction while on high-dose statins. This residual risk is due to the relatively high levels of remnant particles of triglyceride-rich lipoproteins. Their atherogenic effect was best documented by the Copenhagen General Population Study [34]. These remnant particles also carry cholesterol to the arterial wall and are able to furnish it to the monocyte/macrophage lipid rafts and stimulate the inflammatory status of cells (−5). Although it has been suggested that the atherogenic effect of LDL cholesterol is—unlike that of triglyceride-containing lipoproteins—independent of inflammation, it is most likely that both lipoproteins require an inflammatory stimulus to accelerate atherogenesis. We assume that the beneficial dietary modification replacing the presence of palmitate and palmitoleate with a higher proportion of n-3 FAs is able to positively influence the process of atherogenesis and stabilize pre-existing plaques.

The metabolic pathways that influencs the potential synergy of diet-induced decreased lipoprotein concentrations and adipose tissue inflammatory status in CHD mortality are intricate (Figure 5). Decreasing inflow of LDL particles is further potentiated by slowing cholesterol molecule transfer to circulating monocytes which, in turn, translates into slower macrophage polarization to the proinflammatory phenotype [26]. At the same time, an improved proportion of n-3 PUFAs and SAFAs in the cell membrane decreases CHD mortality by diminishing the proportion of PIMs in adipose tissue, slowing down the production of TNF-α and inflammatory interleukins with a beneficial effect on the arterial wall.

As it is, we are unable to explain the unusual proinflammatory effect of palmitoleate on adipose tissue. A plausible and possible explanation is that palmitate is the end product of FA synthesis from a non-lipid substrate and a locally produced saturated molecule, whereby palmitate converts in a one-step process (by stearyl CoA dehydrogenase) to locally acting palmitoleate. However, data regarding palmitoleate local effects are still lacking.

A limitation of this paper is that the initial data and confirmation of the decrease in intravascular cholesterol levels were obtained three decades ago, whereas the above data about inflammation are the result of recent research. It is obvious that only our recent adipose tissue inflammation data put the older epidemiological data into context, enabling a better understanding of the complex issue of decreased cardiovascular mortality. We would like to underline that the above changes clearly show that even “well meant” subsidies may not necessarily have a beneficial impact on society but may be outright counterproductive or even harmful.

## 5. Conclusions

In conclusion, we hypothesize that the sudden economic change led to a substantial diet change and an immediate decrease in atherogenic lipoproteins together with decreased proinflammatory status of adipose tissue. The synergy of both these beneficial changes enhanced the stability of pre-existing coronary plaques in a proportion of the population and immediately reduced CHD mortality.

## Figures and Tables

**Figure 1 biomedicines-10-02871-f001:**
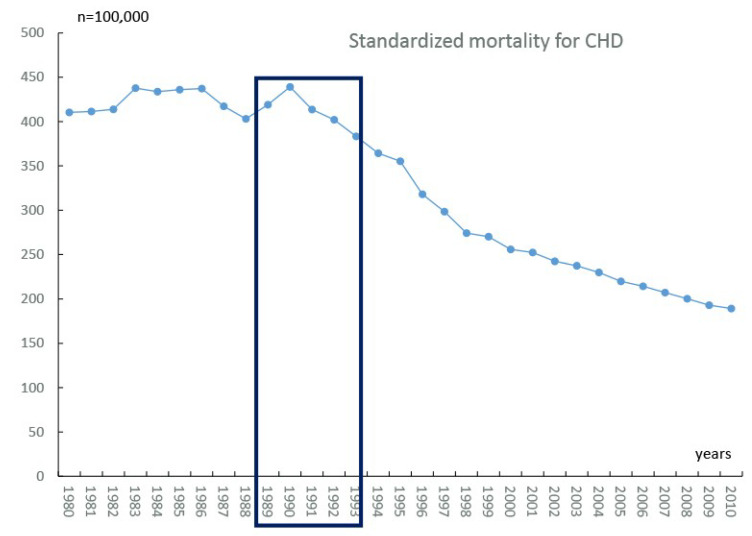
Development of standardized mortality of Czech men/100,000 population.

**Figure 2 biomedicines-10-02871-f002:**
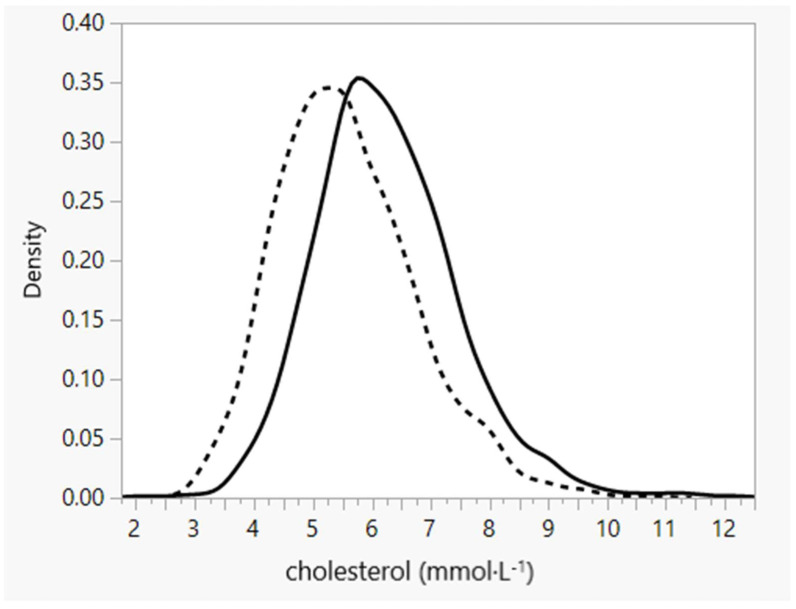
Distribution of total cholesterol levels in a 1% random population sample in 6 districts of the Czech Republic in 1988 (*n* = 2988—solid line) and 1992 (*n* = 2763—dashed line).

**Figure 3 biomedicines-10-02871-f003:**
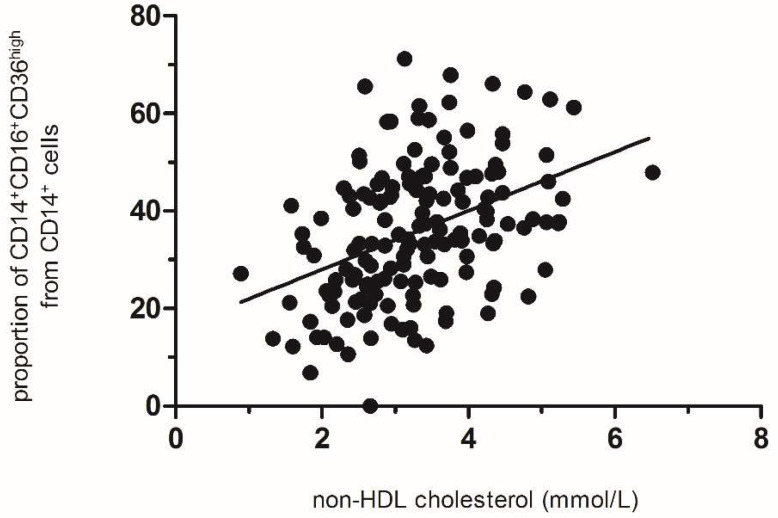
Relation of the proportion of PIMs to non-HDL cholesterol concentration (*n* = 154, *p* < 0.001, r = 0.41). Each dot represents data of one subject (LKD). The line express correlation of the data.

**Figure 4 biomedicines-10-02871-f004:**
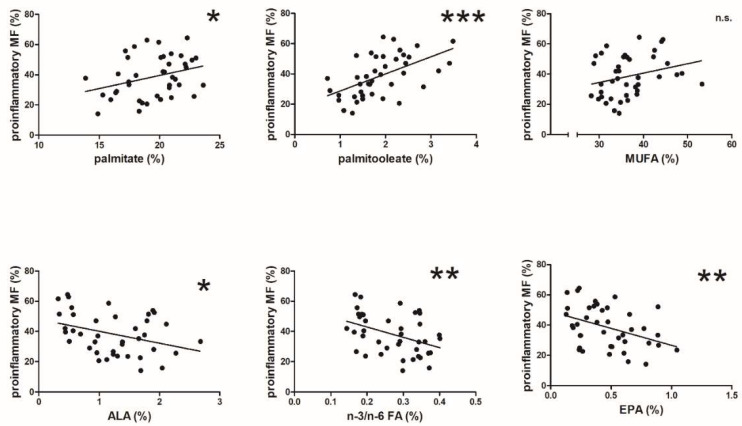
Relation of the proportions of PIMs to those of different FAs of the visceral adipose tissue phospholipids of LKDs (*n* = 43, * *p* < 0.05, ** *p* < 0.01, *** *p* < 0.001). Each dot represents data of one subject (LKD). The line express correlation of the data.

**Figure 5 biomedicines-10-02871-f005:**
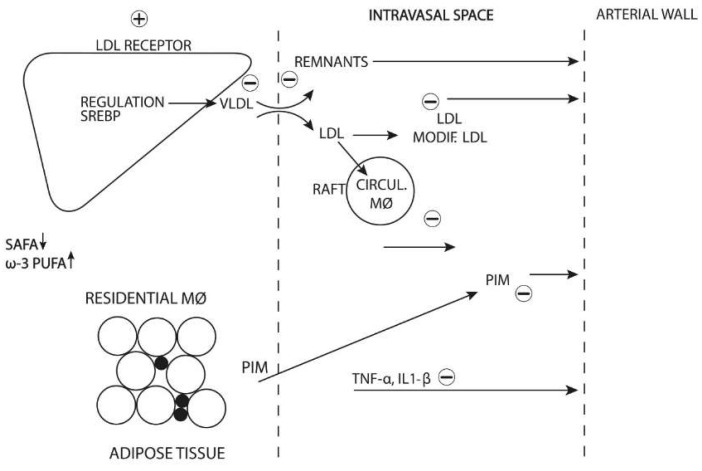
Metabolic changes associated with the decrease in FAs and increase in n-3 PUFA consumption are related to LDL receptors in the liver (mainly) and decreasing of LDL particles in the circulation. At the same time, VLDL synthesis is downregulated and, consequently, the levels of VLDL remnants decline. Changes in SAFAs/n-3 PUFAs decrease PIM polarization in adipose tissue and slow down the production of TNF-α and proinflammatory interleukins. In addition, a decrease in cholesterol levels in the intravascular space impairs its transport to the lipid rafts of circulating monocytes, thus, reducing their inflammation. All these changes attenuate atherogenesis and stabilize pre-existing plaques. All arrows represent metabolites or cells shift. +/− represents increase or decrease of metabolic rate or number.

## Data Availability

The data available from the corresponding author upon request.

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
