# Peer review of "Rapid Drop in Coronary Heart Disease Mortality in Czech Male Population—What Was Actually behind It?"

_biomedicines, 2022, doi:10.3390/biomedicines10112871_

Round 1

Reviewer 1 Report

The authors proved a relationship between cholesterolemia and pro-inflammatory macrophages in adipose tissue and accelerated macrophage polarization with increased palmitate and palmitoleate contents in cell membrane phospholipids.

Major comment

Since the alterations in lipid composition of serum HDL lipoproteins may have a vital role in CHD progression and mortality, could the authors add data concerning that?

Minor comments

Line 219: “2.4. Analysis of fatty acid composition” the authors should refer the biological matrix and the lipids whose the fatty acid composition will be analyzed.

Author Response

Dear reviewer,

thanks for your comments. Please find our answers below:

  • the whole population decrease of non-HDL cholesterol concentration was followed by understandable slight decrease of HDL cholesterol. The total effect was described by a change of total cholesterol/HDL cholesterol ratio which decreased substantialy documenting a decrease of atherogenesis risk. This sentence was included in the Results: Understandably, after a positive dietary change, there was also a slight decrease in HDL cholesterol concentration.
  • As it is described, tissue phospholipids were extracted and FA composition in whole adipose tissue was analyzed. We specified it in the 2.4. paragraph in sentence: The fatty acids in phospholipids of the whole adipose tissue were converted to methyl esters and separated by gas chromatography using a Hewlett-Packard GC system with a flame ionization detector and a carbowax-fused silica capillary column [23]. 
  • Minor corrections of English language will be done by MDPI service.

Reviewer 2 Report

 The authors investigated the mechanism of rapid decline in coronary heart disease mortality in Czech male population in the manuscript entitled “rapid drop in coronary heart disease mortality in Czech male population – what was actually behind it?”. They mentioned that the combination of non-HDL cholesterol drop and the decrease of pro-inflammatory macrophages proportion due to replacement of alimentary fat decreased coronary heart disease mortality immediately. Several concerns have been raised.

1. In the introduction, it would be better to use several paragraphs.

2. In the methods section, it would be better to simply state statistical methodology.

3. In figure 2, which curve is 1988?

4. In figure 3, r-value had better be displayed.

5. In the discussion section, the use of sub-headings is recommended.

Author Response

Dear reviewer,

thanks for your valuable comments and we accepted all your proposals.

  • the introduction was devided into paragraphs
  • description of statistics was included
  • Fig. 2: solid line represents year1988, dashed line is for 1992. The legend of the figure 2 was corrected.  
  • Fig. 3: value of r is included
  • The discussion was devided into paragraphs with sub-headings
  • The English language corrections will be done by MDPI service.